# Power Performance Comparison of SiC-IGBT and Si-IGBT Switches in a Three-Phase Inverter for Aircraft Applications

**DOI:** 10.3390/mi13020313

**Published:** 2022-02-17

**Authors:** Ibrahim A. S. Abdalgader, Sinan Kivrak, Tolga Özer

**Affiliations:** 1Department of Energy Systems Engineering, Yildirim Beyazit University, Ankara 06000, Turkey; 2Department of Electrical, Ostim Technical University, Ankara 06000, Turkey; sinan.kivrak@hotmail.com; 3Department of Electric Electronic Engineering, Afyon Kocatepe University, Afyonkarahisar 03000, Turkey; tolgaozer@aku.edu.tr

**Keywords:** three-phase inverter, high-speed switching, Si-IGBT, SiC-IGBT, microcontroller, aircraft applications

## Abstract

The converters used to integrate the ground power station of planes with the utility grid are generally created with silicon-insulated gate bipolar transistor (Si-IGBT)-based semiconductor technologies. The Si-IGBT switch-based converters are inefficient, oversized, and have trouble achieving pure sine wave voltages requirements. The efficiency of the aircraft ground power units (AGPU) can be increased by replacing existing Si-IGBT transistors with silicon carbide (SiC) IGBTs because of the physical constraints of Si-IGBT switches. The primary purpose of this research was to prove that the efficiency increase could be obtained in the case of using SiC-IGBTs in conventional AGPU systems with the realized experimental studies. In this study, three different experimental systems were discussed for this purpose. The first system was the traditional APGU system. The other two systems were single-phase test (SPT) and three-phase inverter systems, respectively. The SPT system and three-phase inverter systems were designed and implemented to compare and make analyses of Si-IGBTs and SiC-IGBTs performance. The efficiency and detailed hard switching behavior comparison were performed between the 1200-V SiC-IGBT- and 1200-V Si-IGBT-based experimental systems. The APGU system and Si-IGBT modules were examined, the switching characteristic and efficiency of the system were obtained in the first experimental study. The second experimental study was carried out on the SPT system. The single-pulse test system was created using Si-IGBTs and SiC-IGBTs switches in the second experimental system. The third experiment included a three-phase-inverter-based test system. The system was created with Si-IGBTs and SiC-IGBTs to compare the two different switch-based inverters under RL loads. The turning off and turning on processes of the IGBT switches were examined and the results were presented. The Si-IGBT efficiency was 77% experimentally in the SPT experimental system. The efficiency of the third experimental system was increased up to 95% by replacing the old Si transistor with a SiC. The efficiency of the three-phase Si-IGBT-based system was 86% for the six-switch case. The efficiencies of the SiC-IGBT-based system were increased to around 92% in the three-phase inverter system experimentally. The findings of the experimental results demonstrated that the SiC-IGBT had a faster switching speed and a smaller loss than the classical Si-IGBT. As a result of the experimental studies, the efficiency increase that could be obtained in the case of using SiC-IGBTs in conventional AGPU systems was revealed.

## 1. Introduction

Traditional silicon (Si) power inverters make up the majority of high-power converters used to connect planes’ ground power stations to the available electrical grid and provide the necessary power; however, these converters were inefficient, bulky, and struggled to meet the requirements for pure sine wave voltages. The insulated-gate bipolar transistor (IGBT) is an appropriate transistor for medium-frequency high-power fields because it combines the high input impedance of a MOSFET with the high current density of a bipolar device. Si-IGBT switches have a fundamental disadvantage in that they have a lower switching frequency, resulting in greater passive components, weight, and dc-link capacitor volume [1]. In other words, based on those physical characteristics, the Si-IGBT device is already approaching its theoretical limit [2,3].

With a 3.26 eV gap compared to 1.11 eV for Si technology, silicon carbide (SiC) is the next-generation wide-bandgap material. Faster switching speeds reduce switching losses, and a high critical field leads to higher blocking voltage capabilities, short turn-on and turn-off periods that can handle high power, and lower voltage drop [4,5,6]. These varied properties enable the development of extremely efficient SiC switches with high conduction and switching performance [7,8]. Furthermore, SiC-based switches outperform Si-based switches in terms of performance. The efficiency of the new AGPU can be increased by replacing Si devices with SiC devices due to their compact size and suitable weight. Due to the drawbacks of Si-IGBT switches, there is a desire to replace Si-IGBT devices with SiC-IGBT devices for the proposed AGPU.

The first half of an AGPU consists of a six-pulse rectifier bridge and a direct current (DC) bus capacitor, while the second half consists of a three-phase full-bridge inverter (TPFBI) and a transformer that delivers 400 Hz and 208 V alternative current (AC) electricity. The TPFBI is a critical component of the AGPU that should be capable of operating at high voltage. IGBT short circuit protection and an RC snubber circuit should be used to protect TPFBI. To reduce the effect of parasitic capacitance, high dv/dt switching of these SiC-IGBT switches should be handled in the hardware design.

Several articles compared the flipping performance of SiC switches and Si equivalents. G. Wang et al. and Arun Kadavelugu [9,10] determined that the SiC-IGBT behaviors differ significantly from Si-IGBT due to the larger band gap material, greater breakdown field strength, and high-temperature stability. When SiC-IGBT compared to Si-based switches, it was a promising material for achieving high efficiency. A. K. Tripathi and colleagues [11] created a high-frequency isolated DC–DC converter based on a 15 kV SiC-IGBT. Their measurements for the SiC-IGBT revealed a tradeoff between parasitic capacitance, size, and improved performance to reduce current ringing. Gangyao Wang and colleagues revealed [12] that SiC-IGBT devices had ten times higher breakdown electric field strength than Si-IGBT devices. This situation significantly impacted the power utility applications due to lower losses and a higher operating frequency capability. Lubin Han et al. [13] created a SiC-IGBT-based electronic circuit system. Their findings showed that SiC-IGBT had several advantages over Si-IGBT in terms of blocking the voltage, thermal conductivity, and switching speed, whereas the traditional Si-IGBT structure limited the properties of SiC material. A. Kadavelugu et al. [14] designed and demonstrated 15kV SiC-IGBTs based on medium voltage power converters. Their research revealed SiC-IGBT modules designed to reduce the number of devices and simplify converter topologies. According to [15], the equivalent on-resistance of SiC-IGBTs with the same rated voltage was lower than that of Si-IGBTs due to the shorter drift region thickness. Additionally, SiC-IGBTs were more suitable than Si-IGBT for applications requiring high-current operation and voltage. S. Madhusoodhanan et al. [16] compared 12 kV n-type SiC-IGBT with 10 kV SiC-MOSFET and 6.5 kV Si-IGBT based on 3L-NPC VSC. A. Tripathi et al. [17] presented a three-phase dual active bridge isolated DC/DC-converter-based 15 kV SiC-IGBT. S. Madhusoodhanan and colleagues [18] demonstrated 15 kV multilayer converters based on SiC-IGBTs. According to their findings, SiC-IGBT was determined as better than Si-IGBT in several ways.

Fuentes, Carlos D., et al. [19] presented a comparison of two 190 kVA 3-phase 2-level silicon carbide (SiC)- and silicon (Si)-based industrial voltage source converter designs for 690V networks. The SiC-based design demonstrated better performance in terms of its low cost and characteristics. It was understood from previous studies that SiC-IGBT power devices could achieve higher performance and power density than equivalent Si-IGBT power devices. However, it was understood that SiC-IGBT was not used in AGPUs when the literature was examined. At the same time, no analysis was made that includes the advantages of SiC-IGBTs when used in AGPU systems in respect of efficiency and other losses.

The aim of this study was to reveal the effects and advantages of SiC-IGBT elements in terms of efficiency when used in single-phase systems and three-phase inverters. Thus, the efficiency increase that could be achieved in the case of using SiC-IGBTs in APGU-based systems was yet to be demonstrated with experimental data. In this direction, it was aimed to create Si-IGBTs- and SiC-IGBTs-based single-phase and three-phase experimental systems. Hence, all the theoretical and experimental analyses were realized in this direction. SiC-IGBT modules and Si-IGBT devices were compared under the same conditions and similar experimental test systems. The experimental test system was realized with the same gate driver and the same operational settings by using a single-pulse test circuit. The goal of this comparison was to assess their performance and potential clearly. At the same time, it was aimed to determine which type of IGBT module could be employed in AGPU. Thus, the correct technical groundwork was obtained to choose the type of IGBT modules depending on AGPU application requirements.

In this investigation, three experimental systems were discussed and tested. The first experiment contained the AGPU employing only Si-IGBT (CM150DY-24A) switches [20]. The second experiment was realized by creating a Si-IGBT (CM150DY-24A) and SiC-IGBT (APT60GF120JRDQ3) [21] based SPT system. The third experimental study was carried out on the designed three-phase inverter circuit designed with Si-IGBT(CM150DY-24A) and SiC-IGBT (SK25GH063) [22] switches.

The characteristic behavior of SiC-IGBT and Si-IGBT devices were investigated and compared under resistive and RL loads. The advantages of the SiC-IGBT devices were shown under switching times at 100V. Some measurements and explanations were added for the effects of switching on (voltage rise-time) and switching off (voltage rise-time) (voltage fall-time).

The efficiency was boosted by up to 95% by replacing the previous Si-IGBTs with SiC-IGBTs at the third experimental system. The SiC-IGBT had a faster switching speed, and a lower loss than the Si-IGBT was obtained from the experimental results. The remainder of the article is organized as follows: in Section 2, the system analysis is presented. The design and considerations for converters are presented in Section 3. The experimental circuits and test systems are presented in Section 4. Finally, there is a discussion in Section 5, and Section 6 contains the conclusion.

## 2. System Analysis

There were three different systems in this study. The first system was the traditional AGPU system, and the other two systems were designed and created within the scope of this study. The second was a single-phase pulse test (SPT) system, and the third was a three-phase inverter system. Thus, three different experimental studies were carried out in this study. In the first experimental study, the APGU system and Si-IGBT modules were examined, and the efficiency of the system was obtained. The second experimental study was carried out on the SPT system. The operating performances of Si-IGBT and SiC-IGBT modules were compared by creating an SPT experimental system. The third experimental study was carried out on the designed three-phase inverter circuit. The operating characteristics of Si-IGBT and SiC-IGBT modules on the three-phase inverter system were investigated. The switching performance and efficiency of Si-IGBT- and SiC-IGBT-based systems were compared in detail. As a result of the experimental studies, it was revealed that an increase in efficiency could be obtained in the case of using SiC-IGBTs in conventional AGPU systems.

The general features of the AGPU system and the content of the working principle are expressed in Section 4.1. The SPT structure and features of the system are explained only in Section 4.2 since the structure of the SPT system created within the scope of the study was simpler than the three-phase inverter system. Therefore, only the three-phase inverter system circuit structure and working principle are given in detail in the next heading. The control circuit and driver circuit analyses were performed to properly analyze the three-phase inverter system.

### 2.1. Control Circuit Analysis

Control of the systems and data collection are just a few of the areas where microcontroller-based circuits are used today [23,24]. The pulse width modulation (PWM) signals were produced using a high-performance microcontroller control circuit in this study. The switching process of IGBTs was controlled by these signals, which converted the DC to AC as pure sinusoidal wave signals. The DsPIC33FJ32MC204 microcontroller was used, which provided four channels with complementary outputs for controlling the duty cycle of the PWM signals. The output PWM signals were multiplied constantly according to the specified frequency using a logic stage level. Timers were used to set the desired frequency for the PWM signals. The period and the prescaler for the timers were set according to the updated event. The timer was used as an interruption to create a pulse signal at every specific time interval to set the frequency.

The control circuit was designed to control the three-phase inverter circuit. A temperature sensor and current sensor were used in the control system of the three-phase inverter system. At the same time, there was a potentiometer for voltage adjustment of the output signal. The temperature sensor was mounted on the IGBT modules. ACS712 current sensors were connected to each phase leg. Thus, the currents flowing through each phase were measured continuously. The relay connected to the digital pin of the microcontroller was turned off in case the current value exceeded 20 A. A general view of the designed and constructed three-phase inverter system is shown in Figure 1.

#### Development of Software

Eight power PWM outputs are available on the DsPIC33FJ32MC204 microcontroller. Six PWM outputs are required for three-phase SPWM generation. There are two independent outputs, plus their two complements. These signals are used to control S1, S2, S3, S4, S5, and S6, respectively. At the same time, the switching combination of the three-phase inverter can be seen in Table 1. The S1, S3, and S5 switches are independent, while the S2, S4, and S6 switches are complements, respectively.

The sample and steps can be found by the equations:



(1)
$$\mathrm{Sample}=\frac{20\;\mathrm{kHZ}}{400\;\mathrm{HZ}}=50$$





(2)
$$\mathrm{Steps}=\frac{360}{50}=7.2^{0}$$





(3)
$$10000+10000\;\sin\;(x\;\times \;7.2^{0}\times \frac{\pi }{180^{0}})$$



At the start of the program, a lookup table is built, which maintains a fixed number of samples of a sinusoid at a predefined frequency. SPWM output pulses are formed by extracting data from Table 2 in real time using a pointer value and updating the duty cycle registers of the DsPIC33FJ32MC204 microcontroller. There are 25 samples for the half of cycle in the sine lookup table.

The DsPIC-based algorithm developed for the three-phase inverter application is given in Figure 2. The PWM frequency value is set to 5 kHz as the initial setting. All the initial settings are adjusted for ADC, PWM and LCD units. The sinusoidal signal values are stored at the related PWM-based registers. PWMs are then generated according to the switching combination of the six-step three-phase inverter. The control system collects the current and temperature sensor data from the system. The critical current value that could be drawn in the system was determined as 20 A. So, if the current value is under the critical level, the inverter system works. The relay is turned on in case the current value exceeded 20 A. If the temperature of the modules rises above 60 °C, the duty values of the generated PWM signals are set to 0. Thus, the operation of the system is stopped. The potentiometer value is read and PWM duty rates are determined. The amplitude of the output voltage is adjusted by means of the duty rates of IGBTs.

### 2.2. Driver Circuit Analysis

The M81748FP IC was used as the IGBT driver circuit in the three-phase inverter circuit. The gate driver circuit board contains two distinct gate drivers, allowing it to operate both the upper and lower transistors of the half-bridge module simultaneously. The lower gate driver is similar to the upper gate driver in appearance. The stray inductance is minimized due to the proximity of the IGBT module and the driver. Each half-bridge comprises two gate drivers and two separate power sources. One gate driver drives the top switch and the other drives the bottom switch. The generated PWM signals in the microcontroller circuit must be buffered using IGBT drivers. The design of the gate drive considerations was reported in [25]. The circuit diagram of the M81748FP gate driver is shown in Figure 3.

#### 2.2.1. Crosstalk Effect Study

The current flowing through the driver circuit may cause the gate-to-emitter voltage to rise for a short duration. If the gate-to-emitter voltage rises high enough to reach the threshold voltage of the transistor, S_1_ could turn on dangerously. This would cause S_1_ and S_2_ to conduct at the same time, causing a short circuit across the DC bus voltage.

In [26], it was shown that the collector–emitter voltage across the bottom Si-IGBT S_2_ rapidly increased when the top Si-IGBT S_1_ switch turned on and Miller effect current i_Miller_ was produced via Miller capacitance $\;C_{\mathrm{gc}}$. Positive voltage is induced on the gate of lower switch $S_{2}$ by current ${\;i}_{\mathrm{Miller}}$, and the ${\;C}_{\mathrm{ge}}$ at this time can be given as follows:(4)$$V_{\mathrm{ge}}^{+}=+i_{\mathrm{Miller}}\times (R_{\mathrm{driver}}+R_{\mathrm{gin}})+V_{\mathrm{neg}-\mathrm{supply}}$$
where:

$R_{\mathrm{driver}}$ = external driver resistance

$R_{\mathrm{gin}}$ = internal driver resistance

$V_{\mathrm{neg}-\mathrm{supply}}$ = off-state (gate) voltage
(5)$$i_{\mathrm{Miller}}=C_{\mathrm{gc}}\times \frac{{\mathrm{dv}}_{\mathrm{CE}}}{\mathrm{dt}}$$

$C_{\mathrm{ce}}$ = collector–emitter voltage across $S_{2}$

$C_{\mathrm{ge}}$ = gate–emitter voltage

The induced positive voltage could be enough to turn on the lower S_2_ switch incorrectly with the off-state gate voltage of –5V. When the higher switch S_1_ is turn off, the voltage of the collector–emitter across the lower switch S_2_ rapidly drops. Thus, the i_Miller_ current flowing through the gate collector produces Miller capacitance C_gc_. The negative voltage is induced on the gate of the lower switch S_2_ due to i_Miller_.

Furthermore, the gate–emitter voltage can be given by using the equation:(6)$$V_{\mathrm{GE}}^{-}=+i_{\mathrm{Miller}}\times (R_{\mathrm{driver}}+R_{\mathrm{gin}})+V_{\mathrm{neg}-\mathrm{supply}}$$

The negative voltage can exceed the off-state gate voltage to the maximum permitted level. This situation can cause the oxide layer of the gate device to be destroyed and the module to fail. Figure 4 and Figure 5 illustrate the measured gate voltages at the turn-on and turn-off switching times. The gate–emitter negative voltages $V_{\mathrm{ge}}^{-}$ which occurred due to crosstalk effect of lower transistor S_2_ and the gate–emitter positive voltages $V_{\mathrm{ge}}^{+}$ can be seen in these figures.

## 3. Design and Considerations for Converters

The general practical considerations for establishing a converter laboratory setup were discussed in this chapter. The design of gate drivers is one of the topics covered in this chapter.

### 3.1. Circuit for Gate Drivers

Important considerations and needs in gate-driver circuits for SiC-IGBTs have been presented in this section. Considerations such as the needed gate-to-emitter voltage, galvanic isolation, and Miller clamp are covered in the following sections.

#### 3.1.1. Requirements for Gate Drivers

SiC-IGBTs are similar to Si-IGBTs in terms of driving. The SiC-IGBT driver circuit should be low-inductive to reduce the ringing and EMI induced by stray inductance [27]. Another factor to consider is that SiC-IGBT gate drivers are able to withstand a large amount of current. SiC-IGBTs are faster than Si-IGBTs, which means that SiC-IGBTs are more efficient.

The gate-to-emitter voltage of the SiC-IGBT must rapidly increase in order to switch quickly. Therefore, higher gate current is required in order to charge the input capacitance $C_{\mathrm{iss}}$ [28].

The same current capability is required when the IGBT is turned off. The external turn-on and turn-off gate resistors can be reduced to have a larger gate current capability. In addition, the stray inductance of the gate driver must be kept at the minimum in order for the gate current to rise as quickly as desired. SiC-IGBTs require a negative gate-to-emitter voltage such as in Si-IGBTs to achieve a fast and safe turn-off transient. A SiC-IGBT driver typically provides a +20 V positive and −5 V negative voltage for the gate-to-emitter [29].

The undesired out-of-control conduction is prevented by applying −5 V (V_ge(off)_) to the gate pin of the IGBT. Thus, the IGBT enters the cutoff mode quickly because the $C_{\mathrm{iss}}$ input capacitance discharges significantly faster. As a result, the turn-off switching losses are reduced. The SiC MOSFET’s on-state resistance is reduced by the 20 V (V_ge(on)_) highly positive bias voltage. This voltage level provides a smaller turn-on switching loss due to a faster turn-on transient [28].

#### 3.1.2. Signal Supply Galvanic Isolation and DC Power Supply

IGBTs need driver circuits to be controlled smoothly in high-frequency applications. The galvanic isolation is used to provide isolation between the power circuit and the control circuit while designing the driver circuits. The top and bottom transistors in the half bridge topology are controlled by two different gate drivers. The voltage value of the upper transistor can suddenly rise to the source voltage at the moment of switching. This indicates that the reference value of the VGE voltage of the IGBT is variable. It is understood that galvanic isolation is required when applying the control signal to the gate pin of the upper switching element. Fiber optics, optocouplers, and transformers are the three methods for obtaining signal isolation [27].

#### 3.1.3. Miller Clamp

In half-bridge circuit topologies, switching problems may occur due to the operation of SiC-IGBTs at high frequencies. The simultaneous turn on of two transistors can cause a short-circuit condition. At the same time, a short-circuit condition may occur due to a Miller current passing through the gate driver during the turn-off of the transistor, as shown in Figure 6. This is discussed in detail in Section 2.2.1 (Crosstalk Effect Study).

There are several approaches to resolving sudden increase in voltage when the transistor is turned off. One of these solutions is to divide the gate resistor R_g_ into two parts: a turn-on gate resistor $R_{g,\mathrm{on}\;}$ and a turn-off gate resistor $R_{g,\mathrm{off}\;}.$ A lower voltage increase occurs in the gate-to-emitter voltage at the turn-off time of the transistor with the applied of this solution which was presented in the second experiment (SPT) [30].

## 4. Experiments

Two experiments are presented in this section. The AGPU system was used to examine the structure of conventional Si-IGBT-based systems. An experimental study of Si-IGBTs was carried out on this system. The switching and transmission losses of Si-IGBTs were determined and the efficiency of the system was obtained by means of experiments. In the second stage, a three-phase inverter system that could be used in AGPU systems was designed and implemented. Experimental studies based on Si-IGBT and SiC-IGBT were carried out on the created system.

### 4.1. Aircraft Ground Power Units Construction

The AGPU consisted of an input filter inductor (10 kW), six-pulse rectifier thyristor bridge, DC bus capacitor (2 × 10,000 µf, 450${\;V}_{\mathrm{DC}}$), TPFBI, and output filter capacitor, as shown in Figure 7.

The TPFBI consisted of four switches, namely Si-IGBTs (CM150DY-24). Each transistor had a reverse recovery free-wheel diode. The IGBT body diodes (D1 and D2) were attached in parallel with the IGBT within the module to prevent high voltage drops and a super-fast recovery free-wheel diode effect. The key components were the gate driver, DC bus, and output filter. The AGPU cabinet was divided into two parts, as indicated in Figure 8. The first stage employed a rectifier as an AC/DC converter to provide the required DC power, and the TPFBI was utilized later. It transformed power from 380 V (AC) at 50 Hz to 208 V (AC) at a steady frequency 400 Hz. The TPFBI was the central portion of the AGPU and served as an interface between the AC general grid and the AC aircraft loads. This project involved the installation of an AGPU-based TPFBI with Si-IGBT switches. The specifications of the AGPU are presented in Table 3.

#### 4.1.1. The Features of APGU Control System

Eight power PWM outputs are available on the DSPIC30F4011 microcontroller. Four PWM outputs are required for full bridge SPWM generation. There are two independent PWM outputs, plus their two complements. These signals are used to control S1, S2, S3, and S4, respectively, which are illustrated in Figure 7. At the same time, the switching combination of the full bridge can be seen in Table 4. Switches S1 and S3 are independent, while switches S2 and S4 are complements of switches S1 and S3, respectively.

#### 4.1.2. Switching Waveforms and Obtaining Results

In this work, measurements were obtained for an AGPU (10 kVA) utilizing a commercial Si-IGBT power module (1200 V/150 A). The characteristics of the Si-IGBT were determined with different load currents. The measurements in this paper were realized under 450 V DC bus voltage and different RL load currents. The switching times (rise time and fall time), overshoot current, overshoot voltage, and bus voltages with gate resistances (10 Ω) were measured in this experiment.

Voltage and current were measured with an MICsig portable multifunctional oscilloscope (200 MHZ), UT201 clamp multi-meter, and fluke 115 TRUE RMS. The current transformers were put directly on the source terminal to measure the current flowing through the power module. The voltage transformer was used to convert the 120 V to 15 V utilizing a zero-voltage crossing circuit. The Si-IGBT power module was analyzed in order to demonstrate its benefits in terms of high frequency and efficiency [31].

In Figure 9, the switching transient turn-on and turn-off can be observed. The soft recovery action of the diode creates additional losses for the device that the SiC-IGBT would not encounter. On the other hand, all of the SiC-IGBT responses present some sort of ringing. The turn-on and turn-off waveforms of the Si-IGBT were measured under 2.1 A load current conditions. The rise time of the Si-IGBT module was 296.6 ns when the load current was 2.1 A. The fall time of the Si-IGBT module was 753 ns when the load current was 2.1 A. Note that the percentage of losses due to negative overshoot was 11.53%, while positive overshoot accounted for 2.2% of the total switching losses. When the load current (2.1 A) increased, the switching losses also increased, as shown. The switch-off losses were lower than the switch-on losses. Switching on had more oscillations for the Si-IGBT switch.

### 4.2. Single-Pulse Test-Based SiC/Si-IGBT Switches

The second experiment was to build a single-pulse test (SPT) using both single Si-IGBT (CM150DY-24A) and single SiC IGBT (APT60GF120JRDQ3) switches and tested under RL loads (R = 42 Ω, L = 290 uH). The SPT experiment was carried out to acquire and analyze the switching characteristics of both devices. The SPT was divided into two parts, the first of which was the power circuit and the second of which was the control circuit. The power circuit produced −5 V and +15 V, while the control circuit generated the appropriate gate signal to drive both devices.

The SPT activated the IGBT and charged the RL load with current. The SPT should be a broad pulse that charges the load current to the magnitude to be measured. The falling edge and rising edge of a hard-switching transient of the IGBT could be analyzed at the desired current using SPT. Furthermore, the IGBT current magnitude could be adjusted by varying the pulse width of the SPT. Figure 10 shows a Si-IGBT SPT circuit.

#### 4.2.1. Switching Losses and Switching Characteristics

An SPT was used to determine the switching characteristics of the Si-IGBT and SiC-IGBT at 100 V DC input voltages. The original external gate resistances were chosen to achieve the fastest switching. $R_{\left(g,\mathrm{on}\right)}$ was chosen as 10 Ω while $R_{\left(g,\mathrm{off}\right)}$ was 11 Ω. Furthermore, the positive bias voltage was +15 V, and the negative bias voltage was –5V at the gate terminal of both switches in all experiments. The integration feature of the oscilloscope could be used to determine the switching losses of both switches.

##### Turn-Off Switching Characteristics

The turn-off switching characteristics of SiC-IGBT and Si-IGBT devices were examined using SPT. These tests were realized under the 100V voltage value. Figure 11a shows the experiment turn-on and turn-off characteristics for Si-IGBT at 100V (voltage rise time: 262 ns, voltage fall time: 617 ns), while Figure 11b shows the experiment turn-on and turn-off characteristics for SiC-IGBT 100V (voltage rise time: 85 ns, voltage fall time:161 ns).

Table 5 shows the Si-IGBT and SiC-IGBT turn-on and turn-off switching under the RL loads. The turn-off transient was very fast due to stray inductance in the test circuit; thus, it resulted in a significant voltage overshoot and long-lasting ringing. Furthermore, it was clear that when the collector current was increased, the voltage overshot and ringed. Therefore, the switching durations of the SiC-IGBT at 100 V input voltages could be found and compared.Turn-off Delay Time ($t_{d,\mathrm{off}\;}$)During turn-off, the gate-to-emitter voltage drops to 90% of its on-state value and the collector current decreases to 90% of its on-state value, which is known as the turn-off delay time.Current Fall Time ($t_{f}$)During turn-off, the collector current decreases from 90% to 10% of its on-state value, which is known as the current fall time.
(7)$$\mathrm{Total}\;\mathrm{turn}-\mathrm{off}\;(t_{\mathrm{off}})=t_{d,\mathrm{off}\;}+t_{f}$$Voltage Rise Time ($t_{\mathrm{rv}}$)During turn-off, the voltage rise time is the time it takes for the collector-to-emitter voltage to rise from 10% to 90% of its final off-state value.

##### Turn-On Switching Characteristics

The turn-on switching characteristics were investigated in the same manner through SPT. The SiC-IGBT and Si-IGBT turn-on characteristics were investigated at 100 V. Figure 11a shows the turn-on characteristics experiment for Si-IGBT at 100 V (voltage rise time: 262 ns), while Figure 11b shows the turn-on characteristics experiment for SiC-IGBT at 100V (voltage rise time: 85 ns). Figure 11a,b were used for showing the turn-on and turn-off switching characteristics in the same figures. Table 5 shows the Si-IGBT and SiC-IGBT turn-on and turn-off values.


Turn-on delay-time ($t_{d,\mathrm{on}}$): this is the time when the gate-to-emitter voltage reaches 10% of its final value and when the collector current reaches 10% of its maximum value during turn on.Current rise-time ($t_{r}$): the current rise-time is the time it takes for the collector current to rise from 10% to 90% of its final on-state value during turn on.
(8)$$\mathrm{Total}\;\mathrm{turn}-\mathrm{on}\;(t_{\mathrm{on}})=t_{d,\mathrm{on}\;}+t_{r}$$Voltage fall-time ($t_{\mathrm{fv}}$): the voltage fall-time is the time it takes for the collector-to-emitter voltage to drop from 90% to 10% of its off-state value during turn on.

The current overshoot was increased as the turn-on gate resistance decreased. Due to the reduction in turn-on gate resistance, the switching times during turn-on were significantly reduced. As a result, it caused a slight increase in the amplitude of ringing in the current and voltage waveforms.

### 4.3. Three-Phase Inverter Circuit Construction

In this part, two experiments will be carried out. The first experiment uses SiC-IGBT to develop and implement a three-phase inverter, while the second uses Si-IGBT. The control circuit is the same in both experiments, but the power circuit is built separately.

#### 4.3.1. Three-Phase Inverter Circuit Construction with SiC-IGBT

The first experiment was a three-phase inverter based on SiC-IGBTs (SK25GH063). The voltage rating of the module was 600 V, and the current rating was 30 A. Figure 12 shows a schematic circuit for the three-phase inverter, while Figure 13 shows a simplified electrical equivalent of the laboratory-implemented circuit based on SiC-IGBTs.

M81748FP could be used to drive the IGBT switches on the upper and lower sides of a three-leg inverter. The key features of this gate driver include the ability to prevent Miller current. It was discussed in depth in Section 2.2.1 and Section 3.1.3. This gate driver included two gate drivers, allowing it to drive both the upper and lower IGBTs of the half-bridge module at the same time.

##### Switching Waveforms and Results

In this section, measurements are carried out for a three-phase-inverter-based SiC-IGBT power module. The characteristics with pure resistive load (7 Ω) and RL load (R = 42 Ω, L = 290 uH) have been determined. Furthermore, the switching durations (rise-time and fall-time), overshoot current, and overshoot voltage were all measured in this experiment. To measure voltage and current, a MICsig handheld multifunctional oscilloscope (200 MHZ) and Hantek clamp meter were used. Figure 14 shows the gate-to-emitter voltage turn-on and turn-off delay time values of SiC-IGBT under a 2 A load. All the measurements were performed under 100 V. The percentage of losses due to negative overshoot was 0%, while positive overshoot accounted for 4% of the total switching losses for SiC-IGBT. Figure 15 shows the collector-to-emitter voltage characteristics of Si-IGBT at 100 V with a 1A load (rise time: 296 ns, fall time: 753 ns). Figure 16 shows the collector-to-emitter voltage characteristics of SiC-IGBT at 100 V with a 1 A load (rise time: 47 ns, fall time: 36 ns).

#### 4.3.2. Three-Phase Inverter Circuit Construction with Si-IGBT

A three-phase inverter based on Si-IGBTs(CM150DY-24A) was the second experiment in this section. The voltages of the module and current ratings were 1200 V and 150 A, respectively. On both the top and bottom sides of a three-leg inverter, the same gate driver (M81748FP) was utilized to drive both IGBT switches (Si-IGBT and SiC-IGBT). The same control circuit and experimental environment was used as the SiC-IGBT-based system. SiC modules were removed from the three-phase inverter system and replaced with Si-IGBT modules.

##### Switching Waveforms and Results

Measurements for three-phase inverter-based Si-IGBT power modules are performed in this section. The characteristics have been measured under a pure resistive load as well as RL load (R = 42 Ω, L = 290 uH). This experiment also measures the switching durations (rise time and fall time), overshoot current, and overshoot voltage. To measure the voltage and current, the same multifunctional oscilloscope (200 MHZ) and Hantek clamp meter were employed. Figure 17 depicts the Si-IGBT turn-on (261 ns) and turn-off (617 ns) delay time under the 1 A load. All of the tests were carried out at a voltage of 100 volts. The percentage of losses due to negative overshoot was 16% and the positive overshoot caused 5% of the total switching losses for Si-IGBT under a 1 A load.

Table 6 shows the values found during turn-on and turn-off delay times for both Si-IGBT and SiC-IGBT, where the positive and the negative overshoot percentages are also shown.

### 4.4. Total Power Losses and Efficiency

The Si-IGBT and SiC-IGBT should be compared under the identical conditions to assess the efficiency of the three-phase inverter. In this section, the performance of three different switches from various companies was compared. One was a Si-IGBT (CM150DY-24 A) from MITSUBISHI and the other was a SiC-IGBT from Advanced Power Technology (APT60GF120JRDQ3) and SEMIKRON (SK25GH063). The comparison considered conduction and switching losses in the transistors, while the gate driver and diode losses were neglected. Table 7 lists the modules that were chosen. Six experimental data were obtained for comparing the power losses of the switches. Table 8 shows the total power losses and total efficiency and efficiency comparison for RL loads (42 Ω, 290 uH) under 100 V input voltage. These values were handled using four parameters (V_ce_, R_ce(on)_, E_T,on_, and E_T,off_). from the datasheets of devices. Table 9 shows the obtained losses of the experimental tests. These losses were calculated under a 2.1 A load at the same current condition.

#### 4.4.1. Total Switching Power Losses

The switching power losses of a high-frequency switching IGBT can be determined using a three-phase inverter based on both Si-IGBT and SiC-IGBT. It is feasible to study both the turn-on and turn-off transients of the transistor in a three-phase inverter at a given collector-to-emitter voltage and collector current. This allows the overall switching power losses of the transistor to be calculated. The power loss $p_{T,\mathrm{loss}\;}$ is given by:(9)$$P_{T,\mathrm{loss}}\;(t)=V_{\mathrm{ce}}\;(t)\cdot i_{c\;}\;(t)$$
where $v_{\mathrm{ce}\;}$ is the collector-to-emitter voltage and $i_{c\;}$ is the switching transistor’s collector current.

#### 4.4.2. Conduction Power Losses

The on-state collector-to-emitter resistance $R_{\mathrm{ce}\left(\mathrm{on}\right)\;}$ of the transistor causes conduction power losses when the transistor is on.

The power losses due to conduction are calculated as follows:(10)$$P_{T,\mathrm{Cond}}\;(t)=R_{\mathrm{ce}\left(\mathrm{on}\right)}\;(t)\;\times \;i_{c}^{2}\;(t)$$
where $R_{\mathrm{ce}\left(\mathrm{on}\right)}$ is the total collector-to-emitter resistance.

#### 4.4.3. Averaged Switching Losses

The losses associated with changing the state of a component in an inverter from on to off are known as switching losses. During a switching transient, the collector-to-emitter voltage $V_{ce}$ and collector current $I_{c}$ will both be greater than zero and overlap for some period. As a result, there is a power dissipation in the IGBT due to $P_{T,\mathrm{loss}}$ = $V_{\mathrm{ce}}$ ∙ $i_{c}$. During both turn-off and turn-on switching, transistor switching losses occur.

The total switching power loss ($P_{T,\mathrm{sw}}$) in transistor is:(11)$$P_{T,\mathrm{sw}}=\frac{1}{\pi }\;\times \;f_{\mathrm{sw}\;}\;(E_{T,\mathrm{on}}+E_{T,\mathrm{off}})$$

The switching energy losses during turn-on ($E_{T,\mathrm{on}}$) and turn-off ($E_{T,\mathrm{off}}$), respectively, can be found by investigating the power waveform directly. Using the oscilloscope’s integration feature, $E_{T,\mathrm{on}}$ and $E_{T,\mathrm{off}}$ can be found in a three-phase inverter. These energy losses will remain the same independently of the transistor’s switching frequency.

#### 4.4.4. Total Output Power

The three-phase output power of the three-phase inverter is given by:(12)$$P_{\mathrm{out}}=3\;\times \;V_{\mathrm{load}}\;\times \;I_{\mathrm{load}}\;=\;3\;\times \;m\;\times \;\frac{V_{DC}}{2\surd 2}\;\times \;\frac{i_{load}}{\surd 2}$$

#### 4.4.5. Total Power Losses

The total losses of conduction and switching losses in the three-phase inverter are given by:(13)$$P_{T,\mathrm{tot}}=P_{T,\mathrm{Cond}}+P_{T,\mathrm{sw}}$$

#### 4.4.6. Total Losses

The total power losses in three-phase inverter, disregarding gate drivers and diodes, are:(14)$$P_{\mathrm{loss}}=6\;\times \;P_{T,\mathrm{tot}}$$

#### 4.4.7. Total Efficiency

The total efficiency of the power three-phase inverter is calculated by:(15)$$\eta =\frac{P_{\mathrm{out}}}{P_{\mathrm{out}}+P_{\mathrm{loss}}\;}$$

## 5. Discussion

In this research, firstly, the AGPU was developed with a Si-IGBT, and then the experimental AGPU test bench was used to investigate the Si-IGBT characteristics under various RL loads. The purpose of the SPT experiment was to study and analyze the switching characteristics of single Si-IGBT and SiC-IGBT transistors. Furthermore, the characteristics of Si-IGBT and SiC-IGBT were examined and evaluated. Their performance under resistive and RL loads (R = 42 Ω and L = 290 uH) was compared. A review of different articles on more electric aircraft systems (MEA) based on output filters, control types, field applications, semiconductor devices (SiC-MOSFET), and modulation techniques are investigated in [32,33,34,35,36]. At the same time, different articles were investigated about MEA based on semiconductor devices (Si-IGBTs) [37,38,39,40,41,42,43]. Some of these works were used Si-MOSFET [44,45]. It was seen that SiC-IGBT had not been used for AGPU (400 Hz, 208 V) in MEA-based works. The turn-off and turn-on switching times were obtained with ${\;R}_{g,\mathrm{on}\;}$= 10 Ω and ${\;R}_{g,\mathrm{off}\;}\;$= 11 Ω, respectively. The positive overshoot, negative overshoot, and switching times for turning on and off were measured, and data were obtained from the test outcomes. The percentage of losses due to negative overshoot was 16% and, positive overshoot caused 5% of the total switching losses for Si-IGBT. In the same conditions, these values were obtained as 4% and nearly 0%, respectively. It could be understood from these results that SiC-IGBT had less energy loss at switching times.

The Si-IGBT and SiC-IGBT modules were compared regarding the turn-on speed. The SiC-IGBT modules were turned on almost six times faster than Si-IGBT and needed less time to reach a steady-state value. However, SiC-IGBT had more oscillations. SiC-IGBT modules required less time to turn off than Si-IGBT modules due to the absence of tail current. As a result, the necessary technical foundation was provided to choose from depending on the aircraft application requirements. Three critical features could be seen from the waveforms shown in previous figures and tables: large overshoot current, voltage, and current overlap, and voltage drop. Table 8 and Table 9 show the total power losses and total efficiency under 100 V input voltage with RL load with theoretical and experimental, respectively. Theoretically, the efficiency of Si-IGBT switches was 86% in the single-pulse test and 77% in experimental applications. At the same time, the theoretical efficiencies of SiC-IGBT switches were calculated as 96% and 99%, respectively, in the single-pulse test. In real-time application, the efficiencies of the SiC-IGBT-based system were 92% and 95%. In the three-phase system, efficiencies were obtained only experimentally. The efficiency of the three-phase Si-IGBT-based system was 86% for the 6-switch case and 76% for the 12-switch case in the AGPU system. The efficiency of the three-phase SiC-IGBT-based system was obtained as 92% for the six-switched case.

## 6. Conclusions

In this study, it was proven that the efficiency increase could be obtained in the case of using SiC-IGBTs in conventional AGPU systems with the realized experimental studies. Three different experimental systems were discussed in accordance with this purpose.

The Si-IGBT-modules-based APGU system was examined and the efficiency of the system was obtained in the first experimental study. The SPT system was designed and created for the second experimental study. The operating performances of Si-IGBT and SiC-IGBT modules were compared by the created SPT experimental system. The third experimental study was carried out on the designed three-phase inverter circuit. The operating characteristics of Si-IGBT and SiC-IGBT modules on the three-phase inverter system were investigated. The switching performance and efficiency of Si-IGBT- and SiC-IGBT-based systems were compared in detail.

The comparison of the hard-switching behavior of a SiC-IGBT module and a Si-IGBT module was realized under the same layout and identical driving conditions. The explanations were provided regarding the effects of switching characteristics. The collector–emitter voltage, gate–emitter voltage, positive overshoot, negative overshoot, and switching times (rise time and fall time) were measured. The overall switching time was lower since the voltage fall time for the SiC-IGBT was faster than the Si-IGBT, as shown in previous figures and tables. The SiC-IGBT had a higher switching speed and significantly lower loss than Si-IGBT. At the same time, it was obtained that SiC-IGBT could work with high efficiency and high power density. It was then determined that SiC-IGBT modules achieved greater efficiency than Si-IGBTs in the single-pulse test and three-phase-based applications. While the efficiency of Si-IGBT was obtained as at least 77% in single-pulse-test-based experimental applications, it was obtained as at least 92% in SiC-IGBT-based application. Likewise, in the six-switch-based three-phase inverter application, the efficiency of Si-IGBT was at least 86%, while it was at least 92% in the SiC-IGBT-based application. According to the analysis, the AGPU with an Si-IGBT efficiency of only 86% may be achieved at a switching frequency of 20 kHz. Then, the efficiency of the SiC-IGBT could be increased up to 92%. The rate of efficiency could be increased by replacing the old Si transistor with a SiC. The results of the experimental studies prove that the efficiency could be increased in the case of using SiC-IGBTs in conventional AGPU systems.

## Figures and Tables

**Figure 1 micromachines-13-00313-f001:**
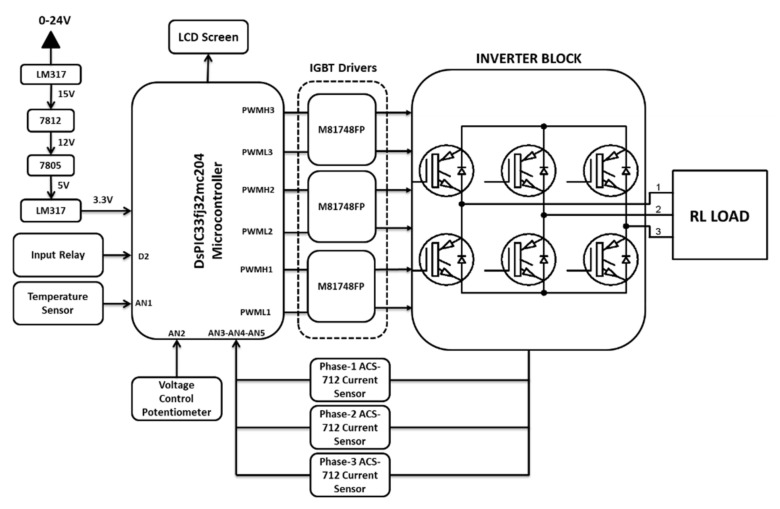
Designed and implemented three−phase inverter system block schema.

**Figure 2 micromachines-13-00313-f002:**
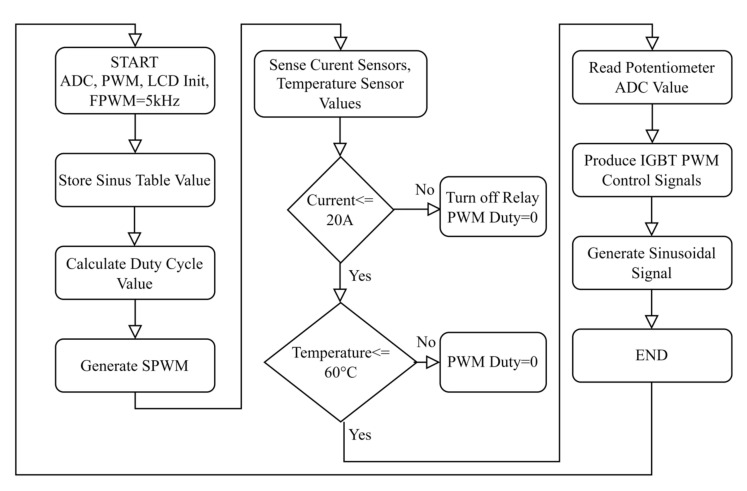
Designed and implemented three−phase inverter system block schema.

**Figure 3 micromachines-13-00313-f003:**
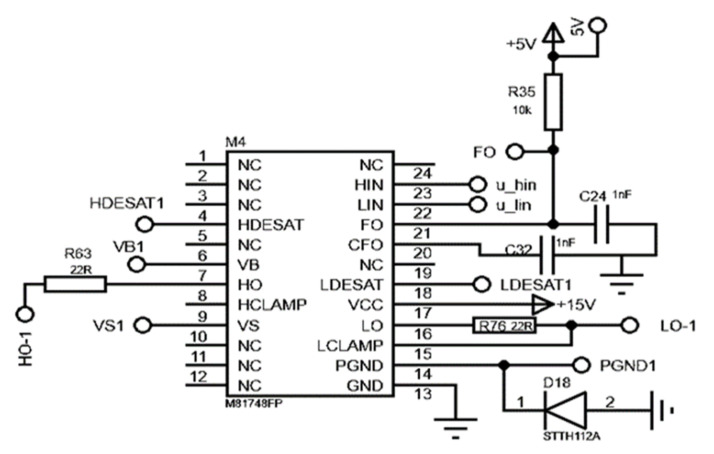
Circuit diagram of the M81748FP gate driver.

**Figure 4 micromachines-13-00313-f004:**
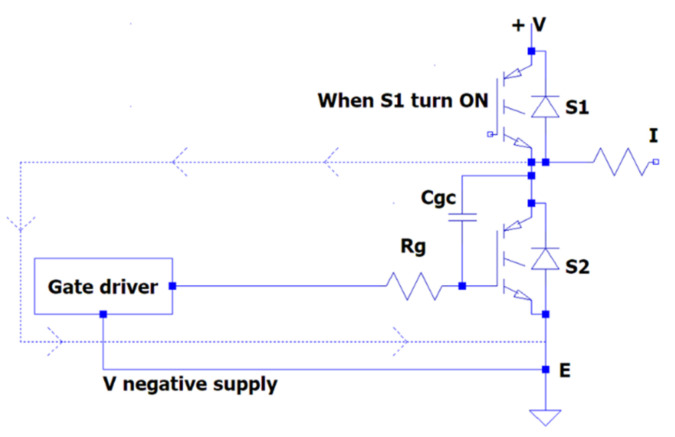
Crosstalk effect when $S_{1}$ is turned on.

**Figure 5 micromachines-13-00313-f005:**
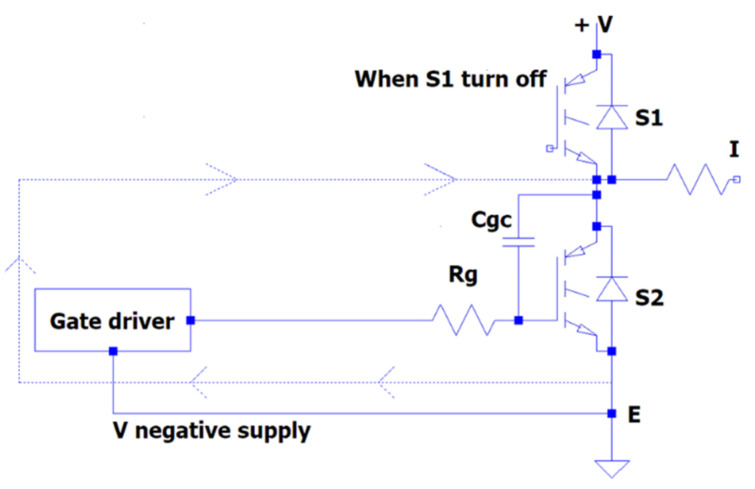
Crosstalk effect when $S_{2}$ is turned off.

**Figure 6 micromachines-13-00313-f006:**
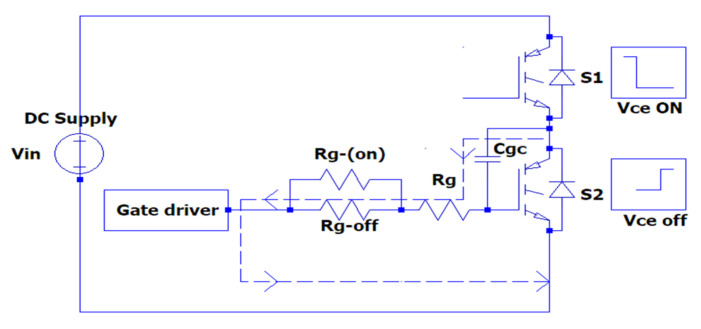
Miller current flowing schema.

**Figure 7 micromachines-13-00313-f007:**
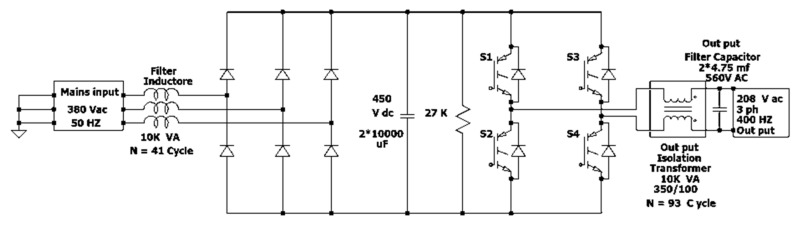
System diagram of single phase for AGPU.

**Figure 8 micromachines-13-00313-f008:**
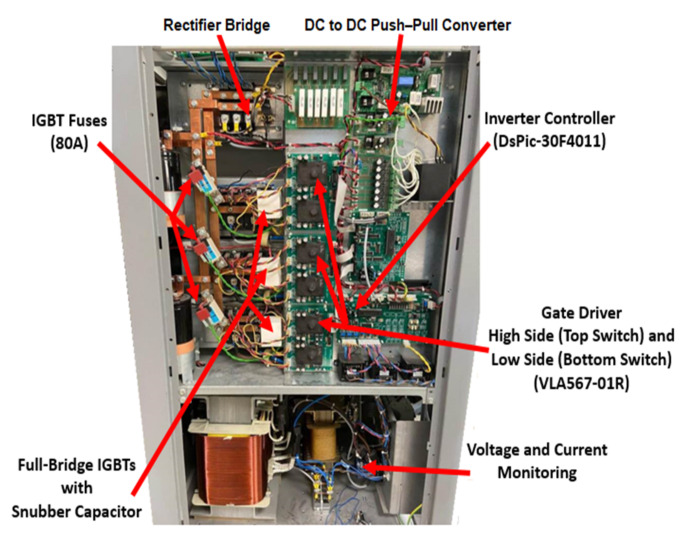
Aircraft ground power units cabinet.

**Figure 9 micromachines-13-00313-f009:**
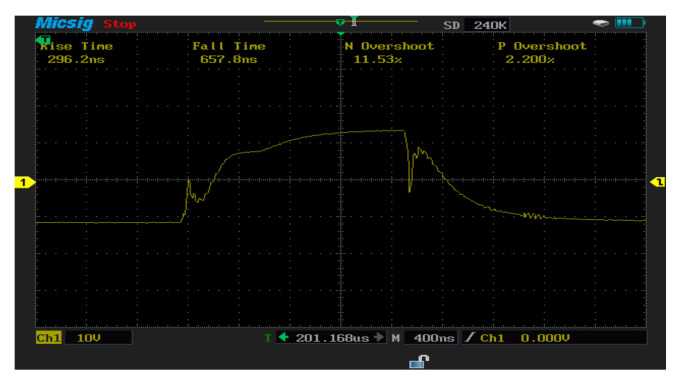
Switching waveforms of Si−IGBT for three−phase aircraft ground power units cabinet.

**Figure 10 micromachines-13-00313-f010:**
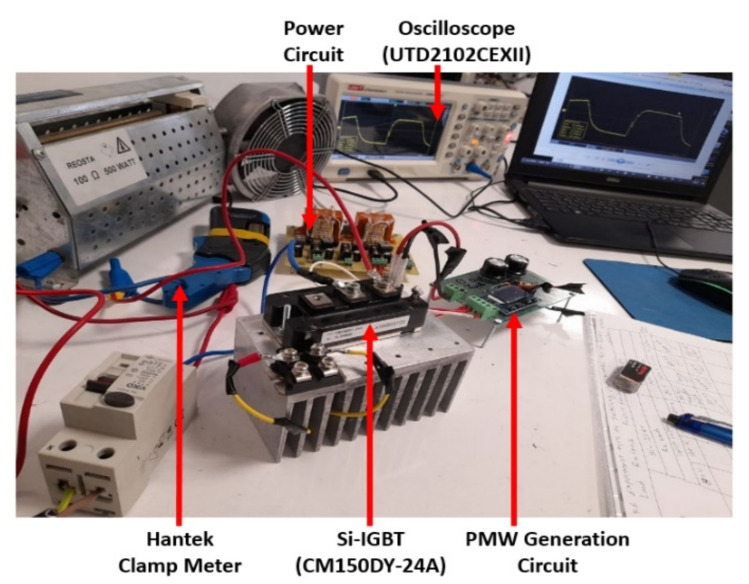
Illustration of the laboratory setting single−pulse test using Si−IGBT (CM150DY−24A) switch.

**Figure 11 micromachines-13-00313-f011:**
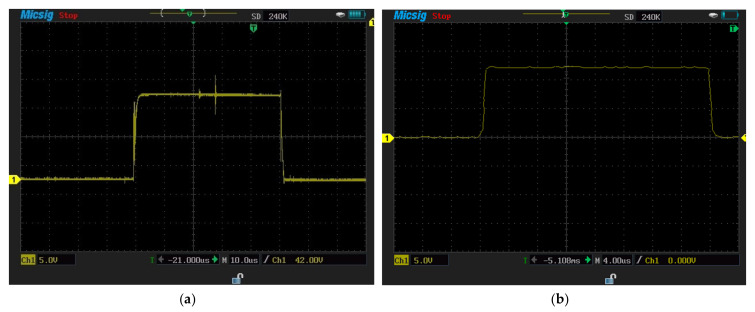
(**a**)**:** Experiment turn−on and −off characteristics for Si−IGBT at 100 V 2 A (voltage rise time (262 ns) and voltage fall time (617 ns)), (**b**): experiment turn−on and −off characteristics for SiC−IGBT at 100 V 2 A (voltage rise time (85.27 ns) and voltage fall time (161 ns)).

**Figure 12 micromachines-13-00313-f012:**
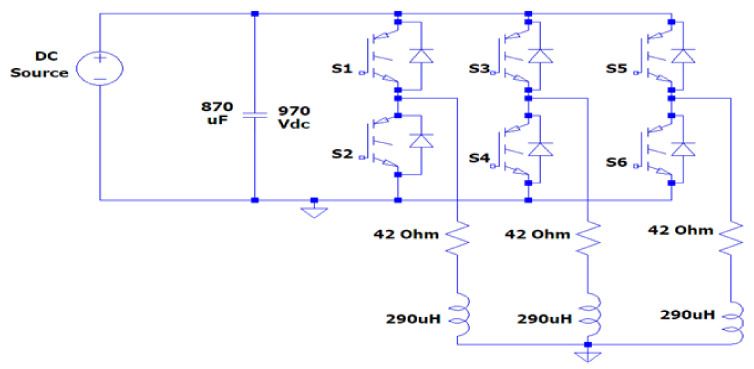
Schematic circuit for three−phase inverter based on SiC−IGBTs.

**Figure 13 micromachines-13-00313-f013:**
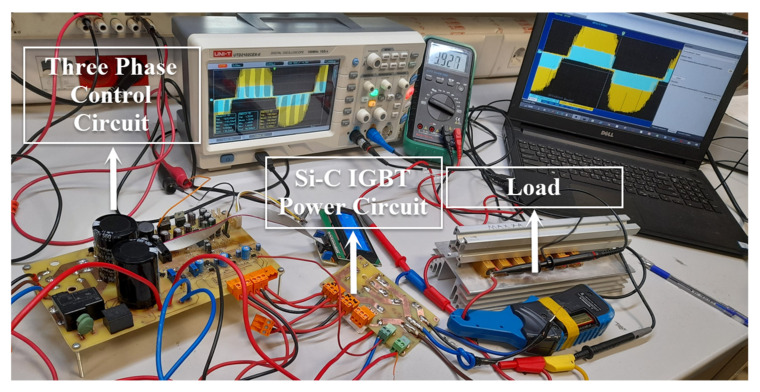
Electrical equivalent of the laboratory circuit three−phase−inverter based SiC−IGBTs.

**Figure 14 micromachines-13-00313-f014:**
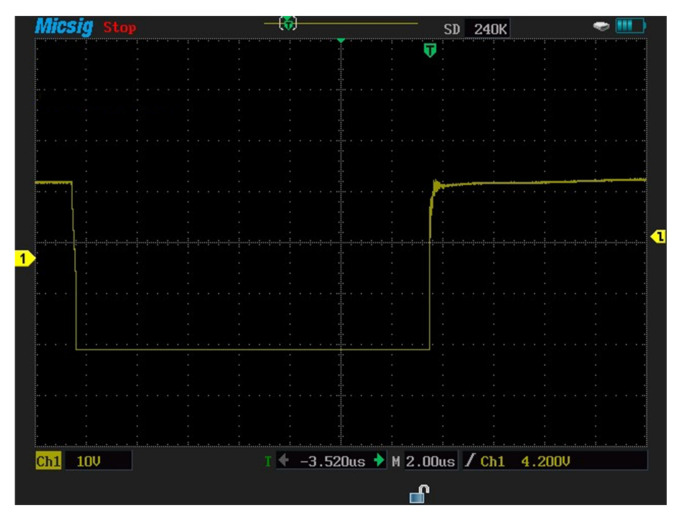
Gate−to−emitter voltage for SiC−IGBT-based three-phase inverter (turn−on: 85 ns, turn−off: 161 ns).

**Figure 15 micromachines-13-00313-f015:**
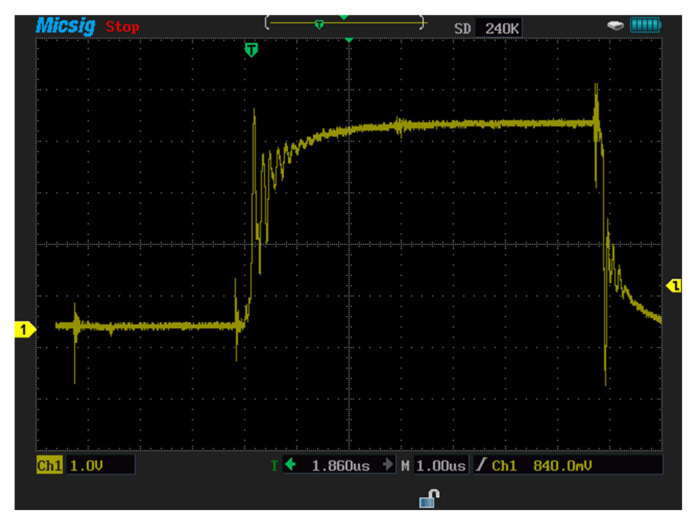
Collector−to−emitter voltage for Si−IGBT-based three-phase inverter (rise time: 296 ns, fall time: 753 ns).

**Figure 16 micromachines-13-00313-f016:**
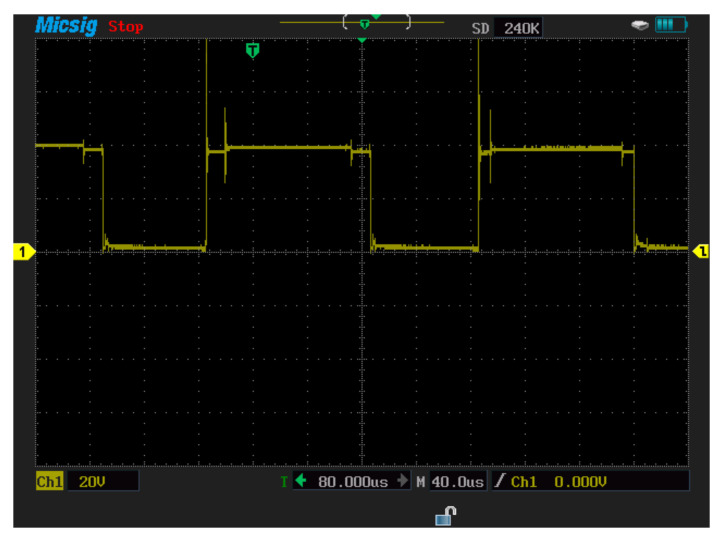
Collector−to−emitter voltage for SiC−IGBT-based three−phase inverter (rise time: 47 ns, fall time: 36 ns).

**Figure 17 micromachines-13-00313-f017:**
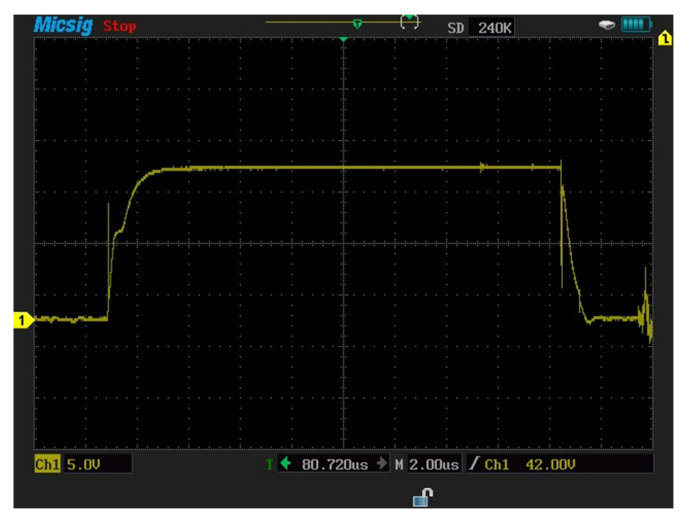
Si−IGBT experiment turn−on (261 ns) and turn−off (617 ns) delay time.

**Table 1 micromachines-13-00313-t001:** Switching combination of six−level three−phase inverter.

Load Line Voltage (V_phase-phase_)	Switching States					
+V_dc_	S1	S2	S3	S4	S5	S6
+V_dc_	1	0	0	0	0	1
0	1	1	0	0	0	0
+V_dc_	0	1	1	0	0	0
−V_dc_	0	0	1	1	0	0
−V_dc_	0	0	0	1	1	0
0	0	0	0	0	1	1

**Table 2 micromachines-13-00313-t002:** Sine lookup PWM table.

K	1pu*sin(φ)	K	1pu*sin(φ)
0	0	13	998
1	125	14	982
2	248	15	951
3	368	16	904
4	481	17	844
5	587	18	770
6	684	19	684
7	770	20	587
8	844	21	481
9	904	22	368
10	951	23	248
11	982	24	125
12	998	25	0

**Table 3 micromachines-13-00313-t003:** Aircraft ground power units cabinet specifications.

Model	Specification
Input voltage	380 VAC (3-phase)
Output voltage	209 VAC (3-phase), stage
Output power	10 kVA
Input frequency	50 Hz
Output frequency	400 Hz
Input DC current	85 A
DC bus	450 VDC
Transistor polarity	Si-IGBT N-channel
Cooling	Forced fan
Output transformer	Galvanic isolation transformer
Microcontroller	Microprocessor dsPIC30F4011

**Table 4 micromachines-13-00313-t004:** Switching combination of full bridge.

	S1	S2	S3	S4
Positive Cycle	PWM	Complement PWM S1	OFF	ON
Negative Cycle	OFF	ON	PWM	Complement PWM S3
ZERO	ON	OFF	ON	OFF
ZERO	OFF	ON	OFF	ON

**Table 5 micromachines-13-00313-t005:** Experiment with the characteristics of turn−on and turn−off switching for Si−IGBT and SiC−IGBT under RL loads at 100V. (R = 42 Ω, L = 290 uH).

Turn-On and Turn-Off	Si-IGBT	SiC-IGBT
	RL Loads	RL Loads
	100V Input Voltage	100V Input Voltage
$\mathrm{Voltage}\;\mathrm{rise}\;\mathrm{time}\;t_{\mathrm{rv}}$(ns)	262 ns	85 ns
$\mathrm{Voltage}\;\mathrm{fall}\;\mathrm{time}\;t_{\mathrm{fv}}$(ns)	617 ns	161 ns

**Table 6 micromachines-13-00313-t006:** Experiment with turn−on and −off delay−time characteristics for Si−IGBT and SiC−IGBT under RL loads. (R = 42 Ω, L = 290 uH).

	Si-IGBT	SiC-IGBT
Turn-on delay time	261 ns	85 ns
Turn-off delay time	617 ns	161 ns
Positive voltage overshoot	5%	4%
Negative voltage overshoot	16%	0%

**Table 7 micromachines-13-00313-t007:** Electrical properties of the three chosen SiC-IGBT and Si-IGBT.

Manufacturer	Device Type and Part Number	Used Experiments	$I_{C}\;(A)$ @ TC 100 °C	$V_{ce}(V)$	$R_{ce\left(on\right)}(m\Omega )$	Turn-on Energy $E_{T,on}(\mathrm{mJ})$	Turn-off Energy $E_{T,off}(\mathrm{mJ})$
Advanced Power Technology	SiC-IGBT APT60GF120JRDQ3	SPT	2.1	3	33 mΩ	14.6	6.5
SEMIKRON	SiC-IGBT SK25GH063	Three-phase inverter system	2.1	2.3	33 mΩ	1.1	0.8
MITSUBISHI	Si-IGBT CM150DY-24A	AGPU, SPT, three-phase inverter system	2.1	2.4	356 mΩ	4	16

**Table 8 micromachines-13-00313-t008:** Total power losses and total efficiency under 100V input voltage with RL loads (R = 42 Ω, L = 290 uH).

Circuit Type	Device Type	Total Switching Power Losses $P_{T,\mathrm{loss}}\;(t)$	Total Conduction Losses $P_{T,\mathrm{Cond}}\;(t)$	Total Switching Losses $P_{T,\mathrm{sw}}$	Total Output Power $P_{\mathrm{out}}$	Total Power Losses $P_{T,\mathrm{tot}}$	Total Losses $P_{\mathrm{loss}}$	Total Efficiency η
Single-pulse test	Si-IGBT CM150DY-24A	5.04 W	1.56 W	31.8 W	222.75 W	33.36 W	33.36 W	86%
Single-pulse test	SiC-IGBT APT60GF120JRDQ3	6.3 W	0.145 W	32.7 W	391.8 W	32.8 W	32.8 W	96%
Single-pulse test	SiC-IGBT SK25GH063	4.83 W	0.145 W	3.01 W	391.8 W	3.19 W	3.19 W	99%

**Table 9 micromachines-13-00313-t009:** Total power losses and total efficiency under 100V input voltage with RL loads (R = 42 Ω, L = 290 uH).

Circuit Type	Total Switching Power Losses $P_{T,\mathrm{loss}}\;(t)$	Total Conduction Losses $P_{T,\mathrm{Cond}}\;(t)$	Total Switching Losses $P_{T,\mathrm{sw}}$	Total output Power $P_{\mathrm{out}}$	Total Power Losses $P_{T,\mathrm{tot}}$	Total Losses $P_{\mathrm{loss}}$	Total Efficiency η
Single-pulse test Si-IGBT	5.04 W	1.56 W	63 W	222.75 W	64.5 W	64.5 W	77 %
Single-pulse test SiC-IGBT	6.3 W	0.145 W	15.7 W	391.86 W	15.84 W	15.8 W	92%
Single-pulse test SiC-IGBT	4.83 W	0.145 W	16.3 W	391.8 W	16.47 W	16.47 W	95 %
Three-phase inverter Si-IGBT	12 W	8.9 W	14.8 W	939 W	23.7 W	142.2 W	For 6-switches 86 % For 12-switches 76% (AGPU System)
Three-phase inverter SiC-IGBT	15 W	0.825 W	12.6 W	933 W	13.4 W	80.55 W	For 6-switches 92 %

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
