# Peer review of "Power Performance Comparison of SiC-IGBT and Si-IGBT Switches in a Three-Phase Inverter for Aircraft Applications"

_micromachines, 2022, doi:10.3390/mi13020313_

Round 1

Reviewer 1 Report

This paper seeks to compare SiC-based insulated gate bipolar transistors to Si-based insulated gate bipolar transistors for applications in aircraft ground power units. In general, the structure of the paper made it very difficult to follow, and it was therefore difficult to properly assess the results. Further, there are a number of places throughout the manuscript that seemed to be missing words or sentences. Please carefully proofread the paper before submission. Aside from that, my suggestions for improving the paper are as follows:

  1. In general, the frequent use of pronouns makes it difficult to determine the subject of any particular sentence.
  2. For the Introduction the objective is unclear: what parameters need to be met? What will improving the efficiency allow for?
  3. Lines 39-40: What does it mean to require "greater passive components"?
  4. In general, there are far too few figures in the early sections of the paper, and far too many in the later sections of the paper.
  5. A schematic or black diagram of the AGPU would aid in understanding lines 52-58.
  6. Schematics would help for understanding sections 2.1 and 2.2.
  7. The Crosstalk Effect study requires more details as to what each of hte components in the circuit do.
  8. In general, it is unclear what the function of the paper is: is it a tutorial to design a AGPU? Is it to understand the benefits of SiC-IGBTs in this process?
  9. Section 3 should be supported by diagrams. It is difficult to follow the description.
  10. Please check spelling of words in the figures ("capacitore", ""transforme")
  11. It is often difficult to read text in figures because of the compressed font.
  12. For essentially all of the figures with plots, the data should be plotted in software instead of including screenshots. It is difficult to interpret the axes of the screenshots and make any kind of comparison. When making a comparison, please plot both sets of data on the same axes.
  13. Lines 312-313: Where is it shown that the switching losses are a function of load current?
  14. What are the purposes of Figures 6-8?
  15. Figure 9A and 9B appear to be nearly the same: are they both necessary?

Reviewer 2 Report

A 10kW AGPU is designed and the hard switching behavior of Si-IGBT and SiC-IGBT is tested by  single pulse test in this paper. However, there are several problems that need to be solved:

  1. The experiment of AGPU is incomplete. The efficiency of AGPU based on Si-IGBT and SiC-IGBT is compared in chapter 6, while there is no experiment result of AGPU based on SiC-IGBT. Besides, readers are interested in the differences between the performance of AGPU based on Si-IGBT and SiC-IGBT. What's the difference between the waveform? What's the difference between the working status of Si-IGBT and SiC-IGBT in AGPU? I think those are more worthy of being tested and discussed.
  2. The circuit of single pulse test is too simple to avoid the influence of parasitic inductance in power loop, control loop and measure loop, which makes the result imprecise. The authors are supposed to design ONE PCB to minimize the influence of parasitic parameters.
  3. It is definitely true that the smaller gate resistances, the smaller switching loss. But the noises of EMI must be considered. Smaller gate resistance usually leads to bad EMC problems.
  4. The authors are supposed to make the figures more standard and easier to read.

Round 2

Reviewer 1 Report

This manuscript has been improved with the revisions.

Reviewer 2 Report

minor english change is required.